# Synergies of integrated pest and pollinator management in avocado farming in East Africa: An ex-ante economic analysis

**Charity Wangithi**[1], **Beatrice W. Muriithi**[1]*, **Gracious Diiro**[1], **Thomas Dubois**[2], **Samira Mohamed**[2], **Michael G. Lattorff**[3¤], **Benignus V. Ngowi**[4], **Elfatih M. Abdel-Rahman**[5], **Mariam Adan**[5], **Menale Kassie**[1]

**1** Social Science and Impact Assessment Unit, International Centre of Insect Physiology and Ecology (*icipe*), Nairobi, Kenya, **2** Plant Health Theme, International Centre of Insect Physiology and Ecology (icipe), Nairobi, Kenya, **3** Environmental Health Theme, International Centre of Insect Physiology and Ecology (icipe), Nairobi, Kenya, **4** Tropical Pesticides Research Institute (TPRI), Arusha, Tanzania, **5** Data Management, Modelling, and Geo-Information (DMMG) Unit, International Centre of Insect Physiology and Ecology (icipe), Nairobi, Kenya

¤ Current address: Department of Chemistry, University of Nairobi, Nairobi, Kenya
* bmuriithi@icipe.org

**Data Availability Statement:** All relevant data are within the manuscript and its Supporting Information files.

## Abstract

Using synthetic pesticides to manage pests can threaten pollination services, affecting the productivity of pollination-dependent crops such as avocado. The need to mitigate this negative externality has led to the emergence of the concept of integrated pest and pollinator management (IPPM) to achieve both pest and pollinator management, leading to complementary or synergistic benefits for yield and quality of the harvest. This paper aims to evaluate the potential economic and welfare impact of IPPM in avocado production systems in Kenya and Tanzania. We utilize both primary and secondary data and employed the economic surplus model. On average the potential economic gain from the adoption of IPPM is US$ 66 million annually in Kenya, with a benefit-cost ratio (BCR) of 13:1, while in Tanzania US$ 1.4 million per year, with a BCR of 34:1. The potential benefits from IPPM intervention gains are expected to reduce the number of poor people in Kenya and Tanzania by 10,464 and 1,255 people per year respectively. The findings conclude that policies that enhance the adoption of IPPM can fast-track economic development and therefore improve the livelihoods of various actors across the avocado value chain.

## 1. Introduction

Avocado is an important crop in generating employment, income, and foreign exchange earnings in many sub-Saharan African (SSA) countries [1]. Kenya ranks second in yield capacity in Africa and sixth in the world (1). The area under avocado cultivation and production volume increased by 42% and 118% between 2005–2014, respectively [2]. The country is also among the top seven exporters of avocado in the world, albeit exporting only 10% of its total avocado production [1]. Avocado exports account for 17% of Kenya's total horticultural exports, contributing US$ 54 million to the Kenyan national gross domestic product (GDP) [3]. In Tanzania, avocado production is also an increasingly important enterprise and is listed by the

**Funding:** D.T., B.W.M, S.M, & M.G.L received the funds. This work received financial support from the German Federal Ministry for Economic Cooperation and Development (BMZ) commissioned and administered through the Deutsche Gesellschaft für Internationale Zusammenarbeit (GIZ) Fund for International Agricultural Research (FIA), grant number 17.7860.4–001; the Norwegian Agency for Development Cooperation, the Section for Research, Innovation, and Higher Education, grant number RAF-3058 KEN-18/0005; the Swedish International Development Cooperation Agency (Sida); the Swiss Agency for Development and Cooperation (SDC); the Federal Democratic Republic of Ethiopia; and the Government of the Republic of Kenya. The views expressed herein do not necessarily reflect the official opinion of the donors. The funders had no role in study design, data collection and analysis, decision to publish, or preparation of the manuscript.

**Competing interests:** The authors have declared that no competing interests exist.

Tanzania Revenue Authority among the top ten export products [4], with annual export revenue of US$ 12.7 million [5].

Despite the impressive growth and economic potential of the avocado sub-sector in Kenya and Tanzania, the production which is dominated by smallholder farmers [6,7], is below both the yield and market potential due to several constraints. Key among them include poor market linkages, pre-and post-harvest losses, pests, and diseases [8]. Pests and diseases, notably tephritid fruit flies (the oriental fruit fly *Bactrocera dorsalis* and *Ceratitis* spp.), the false codling moth *Thaumatotibia leucotreta*, and anthracnose (caused by *Colletotrichum gloeosporioides*) are among the greatest biotic threats affecting avocado yield [6,9]. In addition to yield loss impacted by insects and diseases, over 75% of fruit and vegetable crops grown worldwide are vulnerable to pollination deficit [10], contributing to lower production of the pollination-dependent crops. Using synthetic pesticides to control fruit pests, weeds, and diseases kills insects, including pollinators, thus reducing the productivity of pollination-dependent crops such as avocado [11–13]. Increased use of synthetic pesticides reduces the diversity of avocado flower-visiting insects, resulting in reduced production of avocado fruits [14–16].

Integrated pest management (IPM) is a widely promoted and used decision-based process that optimizes multiple pest control tools tactics in an ecologically and economically sound manner. However, IPM does not take into account the effects on pollination services. To reduce impacts on pollinators and to facilitate synergies between crop pollination and pest control practices, the concept of integrated pest and pollinator management (IPPM) was introduced [17,18]. IPPM consists of combining IPM with tools that safeguard or promote pollination services. In this study, the IPPM package integrates conventional IPM (that is, the use of cultural, mechanical, and biological control options) to suppress avocado pests with supplementation of bees to increase the population of pollinators in the farming systems. The package seeks to exploit synergies between IPM and managed bees to enhance crop productivity, income, nutrition, and food security of smallholder farmers in the region. Integration of IPPM into the avocado farming system increases yields in a more environmentally sustainable manner with less pressure to increase production area and thus higher income to the producers.

While the economic benefits of using IPM and the economic importance of pollination services are well documented [19–23], the potential impact of interacting the two (i.e. IPPM) on crop yield, income, and poverty reduction has not yet been empirically quantified. Previous studies have mainly evaluated the benefits of IPM technologies and beekeeping in isolation [19–23]. Using the case of avocado farming in Kenya and Tanzania, our study contributes to the limited literature on integrating IPM and pollinator conservation by quantifying the potential economic and poverty impacts of IPPM. The findings from this study support the development of strategies and policies for promoting the adoption of IPPM for improved incomes and livelihoods of the avocado value chain actors in Africa.

We use the economic surplus model (ESM), based on experimental, household survey, and secondary data. Unlike previous studies such as [24,25], which rely on expert opinions to compute the maximum adoption rates, we used survey data to reduce the bias in the estimates of the IPPM adoption. Further, we use the economic surplus estimates to evaluate the potential (*ex-ante)* poverty impacts of IPPM adoption. *Ex-ante* impact studies that link economic surplus analysis with poverty analysis include [26–28].

## 2. Materials and methods

### 2.1 Conceptualization

Farmers are economically rational and therefore would adopt IPPM to maximize utility or net returns if the cost of adopting the strategy does not outweigh the benefits. IPPM strategy

enhances farmers' yield due to efficient pollination and minimizes the cost of production. Export market penetration for the IPPM-adopting economies is enhanced due to compliance with global safety standards. The supply curve responds to cost-saving and increasing yields through a parallel shift to the right, with the farmers following a sigmoid adoption path. Both Kenya and Tanzania are open economies, exporting avocado, although their contribution to the international market cannot influence the demand and supply forces. Consequently, IPPM adoption is unlikely to influence world prices. We assume an infinitely elastic demand where the two economies are not in a capacity to significantly influence the world prices through the adoption of IPPM technology. Fig 1 shows how supply responds to the IPPM intervention. The adoption of IPPM induces a downward shift in the supply curve from $S_0$ to $S_1$ through a decrease in the unit cost of production and increased yield. The world prices ($P_W$) are assumed to remain constant and can only rise due to other factors not related to the adoption of IPPM in the two countries.

Initially, producers supply at $Q_{W0}$, and after farmers adopt IPPM, supply rises to $Q_{W1}$, increasing producer surplus through the movement of the supply curve. Producers benefit most since they can sell more at the same price while incurring less cost of production. Consumers in the world market do not benefit from the IPPM interventions since their purchasing capacity is not enhanced. Consumers may however indirectly benefit through better quality and chemical-free avocado fruits due to the reduced use of synthetic pesticides, although our study did not quantify such benefits due to the limitation of data. In the domestic market, producer surplus might rise or shrink; rise since farmers are rational and will tend to sell more to exporters as they are now compliant with phytosanitary standards; or shrink since the export

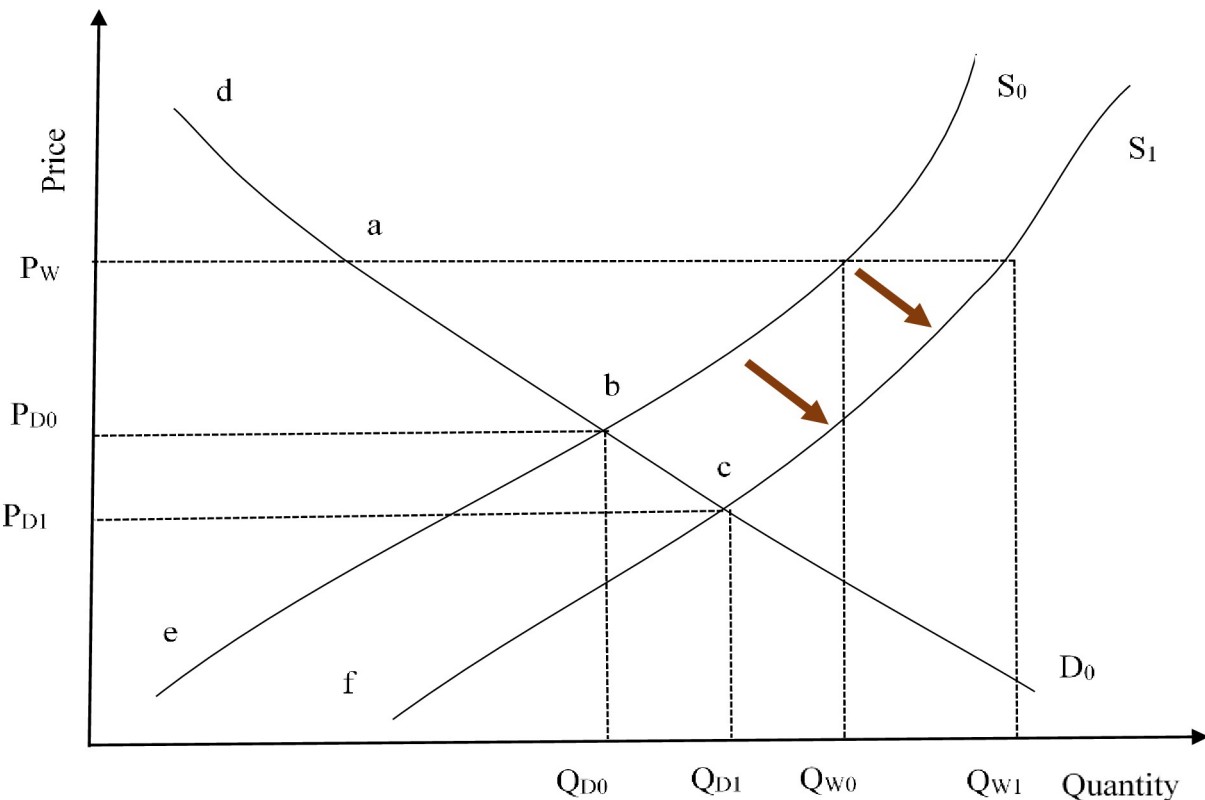

**Fig 1. A conceptualization of induced supply shift due to integrated pest and pollinator management intervention in a small open economy. Source: Adapted from [27].**

market is characterized by grading and sorting; hence the unsuitable fruits are sold domestically at a lower price. Therefore, the domestic producer surplus moves from "$P_{D0}$ b e" to "$P_{D1}$ c f" while consumer surplus moves from "$P_{D0}$ b d" to "$P_{D1}$ c d" as they can now buy more for less.

## 2.2 Estimating economic returns to IPPM intervention

Successful adoption of IPPM may have potential economic gains for avocado farmers and other actors along the value chain. We utilize the economic surplus model (ESM), a widely accepted model for *ex-ante* and *ex-post* impact assessments [23,24,26–30] with the ability to control both international prices and distributional effects [24,29,31]. The introduction of IPPM is expected to cause a direct effect on producers through increased productivity (induced by the availability of pollination services) and reduced production costs (due to a decrease in expenditure on pesticides). The technology-induced shift of the supply curve is estimated as an intercept change (*K factor*) (25–29), which represents the proportionate shift in the supply curve or the per-unit production cost reduction due to the intervention. The K-shift parameter is defined as:

$$K = \left( \frac{ATT_m}{\varepsilon} - \frac{ATT_c}{1 + ATT_m} \right) \times A \qquad (1)$$

where the index $ATT_m$ is the proportionate change in yield of avocado fruits (m) and $ATT_c$ the proportionate change in the cost of avocado production due to the adoption of IPPM, both obtained from the literature (Table 1). A represents the estimated adoption rate of IPPM from the baseline survey data in Kenya and Tanzania, while $\varepsilon$ is the price elasticity of supply also obtained from existing literature. The impact of adopting a technological intervention like IPPM on the economic surplus in a region depends on its openness to trade [28]. Kenya and Tanzania are open economies with 2.6% and 0.4% market share in the global avocado trade, respectively [32]. Therefore, the world prices are held fixed, hence the benefits from IPPM will only accrue to producers and none to the consumers. This is because the adoption of IPPM does not influence open market prices, as shown later in Eq (5). Ideally, annual changes in consumer surplus (CS) and producer surplus (PS) at time *t* are derived as;

$$\Delta CS_t = P_0 Q_0 Z (1 + 0.5 Z_\eta) \qquad (2)$$

$$\Delta PS_t = P_0 Q_0 (K - Z)(1 + 0.5 Z_\eta) \qquad (3)$$

**Table 1. Parameters for the economic surplus model used to estimate the impact of integrated pest and pollinator management in Kenya and Tanzania.**

| Parameter | Data | Source |
|---|---|---|
| The elasticity of supply ($\varepsilon$) | 0.8 | [27,40] |
| Elasticity of demand | -0.2 | [39,46] |
| Expected yield gain IPM | 40.9% | [43,44,47] |
| Expected yield gain pollinators | 20.7% | [14,48] |
| Expected cost reduction-IPM | 30% | [17,43] |
| Probability of success | 70% (60% Tanzania) | [41], household survey data |
| Real discount rate | 8% (11% Tanzania) | Commercial Banks (2020) |
| AgriGDP | US$ 1932 million (Kenya) US$ 1533 million (Tanzania) | [32] |

$$\Delta TS_t = P_0 Q_0 K (1 + 0.5 Z_\eta) \tag{4}$$

However, both Kenya and Tanzania are open economies, therefore;

$$\Delta PS_t = P_W Q_0 K (1 + 0.5 K_\varepsilon) \tag{5}$$

$$\Delta TS_t = \Delta PS_t \tag{6}$$

$P_0$ and $Q_0$ are the price and quantity of avocado production before the introduction of IPPM, $\eta$ is the elasticity of demand, while $Z$ is the relative price change. Parameters described in Eqs (1–6) are presented in Table 1 and discussed in Section 3.

## 2.3 Estimating the impact of IPPM intervention on poverty reduction

Economic growth is the most instrumental component in poverty reduction and in improving the welfare of the people. IPPM is hypothesized to contribute to additional income through increased production, employment creation, and wage effects in the avocado and other sub-sectors through production, consumption, and savings pathways [27,33]. The estimated economic surplus, which is the additional monetary gain to the country's economy due to IPPM intervention, may reduce poverty. Following [27], we estimate the impact of IPPM on poverty reduction as illustrated below;

$$\Delta N = \left[ \frac{\Delta TS}{AgriGDP} \times \delta \right] \times N \tag{7}$$

where $\Delta N$ is the number of people escaping poverty due to changes in the economic surplus, $\Delta TS$ is the change in total economic surplus due to IPPM intervention, AgriGDP is the value of agricultural GDP, $\delta$ is the elasticity of poverty in response to Agricultural GDP and $N$ is the number of poor people in the country.

Both AgriGDP and the number of poor people were obtained from the World Bank [34]. We also calculated the elasticity of poverty from the World Bank data, which compared well with reported figures by Diao et al. [35]. Agriculture is the major contributor to both countries' GDP, creating more than 70% of rural employment [36]. Therefore, an increment in avocado yield and foreign exchange earnings through IPPM intervention is assumed to change GDP and the number of people escaping poverty. In Kenya, a 1% increase in GDP driven by horticultural crops was found to lead to a 1.2% reduction in the country's poverty headcount rate per year. In comparison, Tanzania's poverty rate is more responsive to change in GPD, with a 1% increase in GDP resulting in a 5.1% reduction in the country's poverty headcount rate per year.

## 2.4 Data sources, parameter estimation, and assumptions

We utilize data collected in February and May 2019 from smallholder farmers in selected sites in Kenya and Tanzania where avocado production is predominant. These are Murang'a County in Kenya and Kilimanjaro Region in Tanzania. The two are the leading avocado-producing regions in their respective countries, with Murang'a County accounting for 46.9% of the total avocado production in Kenya [3], while Kilimanjaro Region, specifically Siha District accounts for about 54% of the avocado production in Tanzania [6,37]. Using a multi-stage sampling procedure, a random sample of 410 [38] and 420 avocado-growing households were

selected for the baseline survey in Kenya and Tanzania respectively. To assess the potential adoption of IPPM, their preferences for either the innovation in isolation (i.e. IPM or pollinator supplementation (managed Western honey bee (*Apis mellifera*) hives) or integration of both (i.e. IPPM), were first determined [38]. Upon selecting the most preferred option, the respondents' willingness to buy the selected option based on the current cost of pesticides for the management of the target pests and disease was elicited. A further question on how long it would take them to adopt IPPM informed this study's adoption data.

A log transformation revealed maximum adoption levels of 62.4%, 71.1%, and 68.3% for IPM, pollinator supplementation, and IPPM in Tanzania, respectively, and 53.5%, 83.3%, and 85.9% for IPM, pollinator supplementation, and IPPM in Kenya, respectively. A simulation period of 15 years (2019–2033) was adopted to allow for the research period and adequate time for the pollinator populations to regain efficient pollination services. Further, the research period allowed adequate time to have new entrants into the avocado sector since avocado is a perennial crop where 3-5-year-old trees yield 300–400 kgs (30,000–40,000 fruits)/ha while trees older than 5 years yield 800–1000 kgs (80,000–100,000 fruits)/ha [39]. A research lag of 2 years was considered for this study. Most of the IPM components are available from the market, and therefore scaling will mostly involve farmer training and pilot trials, while beehives of *A. mellifera* are available from local commercial sources.

Production and price statistics were adopted from FAOSTAT and Horticultural Crops Directorate (HCD) (Nairobi, Kenya), the National Bureau of Statistics (Dodoma, Tanzania), the UN Comtrade, and the International Trade Centre [2,3,5,32,40]. Production and consumption growth rates were projected from an average trade and production data computation. Data on the price elasticity of supply and demand was adopted from previous related work (see Table 1). The demand elasticity of -0.2 was adopted from [41], who studied price elasticities of demand for fresh Hass avocadoes in the USA. Since avocado is a perennial crop, supply is inelastic to change in prices during the short run but can be elastic in the long run. Crop-specific acreage elasticities of agricultural supply response to prices in developing countries vary from 0 to 0.8 in the short-run and from 0.3 to 1.2 in the long run for a wide variety of crops [29,42]. Our study, therefore, adopted an elasticity of supply of 0.8 [29].

In *ex-ante* analysis, the probability of research success is mainly derived from experts' opinions especially the scientists involved in the design of the intervention [24,25,27]. However, the probability of success of experts' opinions could be subjective since they may not reflect the current farmers' practices and farming systems. For instance, [25] set 90% as the probability of success based on scientists' opinions from the research program and defined the probability of research success as the likelihood that the biological agents would be successfully identified, bred, and released, and they in turn successfully suppress the coconut mite. Although the scientist had completely identified and multiplied the predators for release, the intensity at which the biological control agent would completely spread throughout the area infested with coconut mites was unclear and only validated through literature which could vary with different agroecological zones. To circumvent this limitation, we use primary data to derive the probability of IPPM research success following [43] using anchored scales which is one of the most efficient and reliable methods. The method, however, was designed to assess business projects instead of farm-related projects. We, thus, modified the individual measurement questions and generated scales from responses to farmers' perceptions and willingness-to-use IPPM. The probability of successful IPPM adoption was 70% in Kenya and 60% in Tanzania. The average response to farmers' willingness to adopt IPPM was less in Tanzania, with the knowledge and perception indicators on IPM and pollinator supplementation lower than those in Kenya.

IPM has been proven to contribute to yield and net income by reducing crop loss and increasing the resilience of the cropping system while lowering the cost of production due to less use of synthetic pesticides [44]. Following [45,46], we adopted a 40.9% yield change due to IPM and a 30% potential cost reduction due to reduced expenditure on pesticide use [19,45,47]. This data on cost reduction and yield increment was used to calculate the *K factor* shift in supply after comparison of how much farmers were spending on pesticides and the potential price of IPPM plus the projected adoption pattern [45]. We adopted a 20.7% yield change due to pollinator supplementation following [14] a study on avocado pollination deficit. The pollinator supplementation scenario however does not have any cost changes. We aggregate the cost and yield changes from the IPM and pollinator supplementation scenarios to build on the IPPM scenario where a potential yield change of 61.6% and 30% potential cost reduction were adopted. The cost of research and development, dissemination, and extension were computed from project documents. IPM, pollinator supplementation, and IPPM would cost US$ 148.3, 167.7, and 345.9 per hectare respectively. Most IPM packages are seasonal (per year), while beehive and colony lifespan is about 3.5 years.

## 2.5 Ethics statement

The study received ethical clearance from the International Centre for Insect Physiology and Ecology (*icipe)* science committee. For the primary data, oral consent was requested from the survey respondents after providing them with a detailed background of the study to allow them to make an informed decision on their participation in the survey.

## 3. Results and discussion

### 3.1 Potential economic returns to IPPM intervention

The results of the economic analysis of integrating IPM with pollinator supplementation over 15 years (2019–2033) and a real discount rate of 8% in Kenya and 11% in Tanzania are presented in Tables 2 and 3 respectively. Producers' surplus equals total surplus since there are no monetary benefits earned by consumers in a small open economy and with the assumed scenario parameters as explained in the previous section.

In Kenya, the total present value (total economic surplus minus discounted research cost) is US$ 222 million for IPM, US$ 446 million for pollinator supplementation, and US$ 996 million for IPPM (Table 2). On average, farmers adopting IPPM would gain US$ 66 million annually in Kenya. The benefit-cost ratio and the IRR of the three simulation scenarios also show the feasibility of the interventions with IPPM having the highest returns. IPPM yields more benefits than adopting either IPM and pollinator supplementation in isolation, emphasizing the need to integrate sustainable pest management and pollinator management. Economic impacts of IPM such as those reported by [19,46] could be enhanced by incorporating pollinator conservation in the upscaling efforts.

Tables 2 and 3 further show the distribution of economic changes in producer surplus through the simulated period. Assuming the world prices remain constant and only rise due to other factors such as enhanced consumer-buying potential that would increase demand, the prices act as an incentive to enlarge their avocado orchards. In the short run (~4 years), farmers enjoy the benefits mainly due to minimized production cost, better pest management, and enhanced pollination. In the long run (>5 years), the new orchards are in production, combining effective pollination and sustainable pest management strategies, resulting in increased avocado yield. The economic gains due to pollinator supplementation and IPPM are conservative and could be underestimated since the economic benefits from such innovations are coupled with other positive externalities, including reduced health and environmental risks, and

**Table 2. Total economic benefits of adopting IPM, pollinator supplementation, and IPPM in avocado production in Kenya in ('Million US$).**

| Intervention | IPM | Pollinator supplementation | IPPM |
|---|---|---|---|
| Year | PS/TS | PS/TS | PS/TS |
| 2019 | 0 | 0 | 0 |
| 2020 | 0 | 0 | 0 |
| 2021 | 1 | 2 | 4 |
| 2022 | 2 | 4 | 9 |
| 2023 | 4 | 9 | 18 |
| 2024 | 7 | 17 | 37 |
| 2025 | 13 | 32 | 69 |
| 2026 | 22 | 55 | 119 |
| 2027 | 34 | 83 | 183 |
| 2028 | 49 | 111 | 247 |
| 2029 | 65 | 133 | 298 |
| 2030 | 78 | 148 | 333 |
| 2031 | 87 | 158 | 357 |
| 2032 | 94 | 165 | 372 |
| 2033 | 99 | 170 | 384 |
| NPV | 222 | 446 | 996 |
| B/C | 6.6 | 10.5 | 12.9 |
| IRR | 52.8% | 68.2% | 76.2% |

*Note*: PS, and TS are producer, and total surplus respectively; B/C is the benefit over cost, IRR is the internal rate of return. NPV is the net present value. Consumer surplus (CS) is zero equating PS to TS.

**Table 3. Total economic benefits of adopting IPM, pollinator supplementation and IPPM in avocado production in Tanzania ('Million US$).**

| Intervention | IPM | Pollinator supplementation | IPPM |
|---|---|---|---|
| Year | PS/TS | PS/TS | PS/TS |
| 2019 | 0 | 0 | 0 |
| 2020 | 0 | 0 | 0 |
| 2021 | 0.0001 | 0.0001 | 0.0002 |
| 2022 | 0.0001 | 0.0002 | 0.0003 |
| 2023 | 0.0003 | 0.0003 | 0.0007 |
| 2024 | 0.0005 | 0.0007 | 0.0013 |
| 2025 | 0.0010 | 0.0012 | 0.0025 |
| 2026 | 0.0018 | 0.0021 | 0.0043 |
| 2027 | 0.0027 | 0.0032 | 0.0065 |
| 2028 | 0.0036 | 0.0043 | 0.0088 |
| 2029 | 0.0043 | 0.0051 | 0.0106 |
| 2030 | 0.0047 | 0.0057 | 0.0118 |
| 2031 | 0.0050 | 0.0061 | 0.0126 |
| 2032 | 0.0053 | 0.0064 | 0.0131 |
| 2033 | 0.0054 | 0.0066 | 0.0136 |
| NPV | 0.0105 | 0.0127 | 0.0260 |
| B/C | 12.40 | 16.57 | 33.94 |
| IRR | 54.9% | 61.1% | 81.3% |

*Note*: PS, and TS are producer and total surplus respectively; B/C is the benefit over cost, IRR is the internal rate of return. NPV is the net present value. Consumer surplus (CS) is zero equating PS to TS.

higher household income through the sale of honey and other bee products, which are beyond the scope of the current study.

Successful implementation and adoption of IPPM in Tanzania will enable the economy to add up to US$ 25,970,600 over the simulated period of 15 years (Table 3). The discount rate in Tanzania is relatively higher than in Kenya based on the higher current cost of capital in Tanzania. The discount rates were also adjusted to inflationary rate changes and compared well with the cost of capital for long-term investment of closely related farm products. Farmers benefit from the rising production volume while prices remain a factor under control by economies other than the adoption of IPPM. Like the Kenyan economy, avocado production and consumption growth rate without IPM, pollinator supplementation, and IPPM intervention would also be realized due to factors other than the intervention.

Table 4 shows changes in quantity produced after the different interventions in Kenya. The change in quantity produced without intervention is only attributable to other factors such as the introduction of new technological innovation, government incentives (e.g. subsidies to encourage production), and relief of import duties enhancing trade and encouraging producers to raise their production levels. Change in consumed quantity also rises due to factors other than IPPM in a small, closed economy such as enhanced purchasing power, change in consumer tastes and preferences towards avocado due to increased awareness of the fruits' nutritional benefits.

A comparison of changes in producer surplus due to IPPM between the two countries over time is shown in Fig 2. A steady increase is observed in the Kenyan economy compared to Tanzania, which corroborates with the current production. Furthermore, the avocado export trade in Kenya is growing exponentially, especially with the recent trade agreements. For instance, the agreement between Kenya and China that was effected in 2019 and a recent one between Kenya and South Korea that took effect in March 2022 (businessdailyafrica.com). In Tanzania, the avocado sector is also growing albeit indolently with contract farming becoming a major business strategy among the two largest avocado producers in the country.

**Table 4. Simulated changes in quantity produced under the IPM, pollinator supplementation, and IPPM interventions in Kenya ('000 tonnes).**

| Intervention year | IPM | Pollinator supplementation | IPPM | No R/D |
|---|---|---|---|---|
| 2019 | 364.9 | 364.9 | 364.9 | 364.9 |
| 2020 | 372.2 | 372.2 | 372.2 | 372.2 |
| 2021 | 379.9 | 380.2 | 380.8 | 379.6 |
| 2022 | 387.8 | 388.4 | 389.7 | 387.2 |
| 2023 | 396.0 | 397.4 | 400.1 | 395.0 |
| 2024 | 404.8 | 407.6 | 413.0 | 402.9 |
| 2025 | 414.4 | 419.7 | 429.8 | 410.9 |
| 2026 | 425.1 | 434.2 | 451.4 | 419.2 |
| 2027 | 437.0 | 450.2 | 476.1 | 427.5 |
| 2028 | 449.7 | 466.1 | 500.5 | 436.1 |
| 2029 | 462.6 | 480.7 | 521.7 | 444.8 |
| 2030 | 475.0 | 493.5 | 539.0 | 453.7 |
| 2031 | 486.6 | 505.2 | 553.7 | 462.8 |
| 2032 | 497.8 | 516.3 | 566.8 | 472.0 |
| 2033 | 508.5 | 527.1 | 579.2 | 481.5 |

Note: R/D refers to the Research and Development.

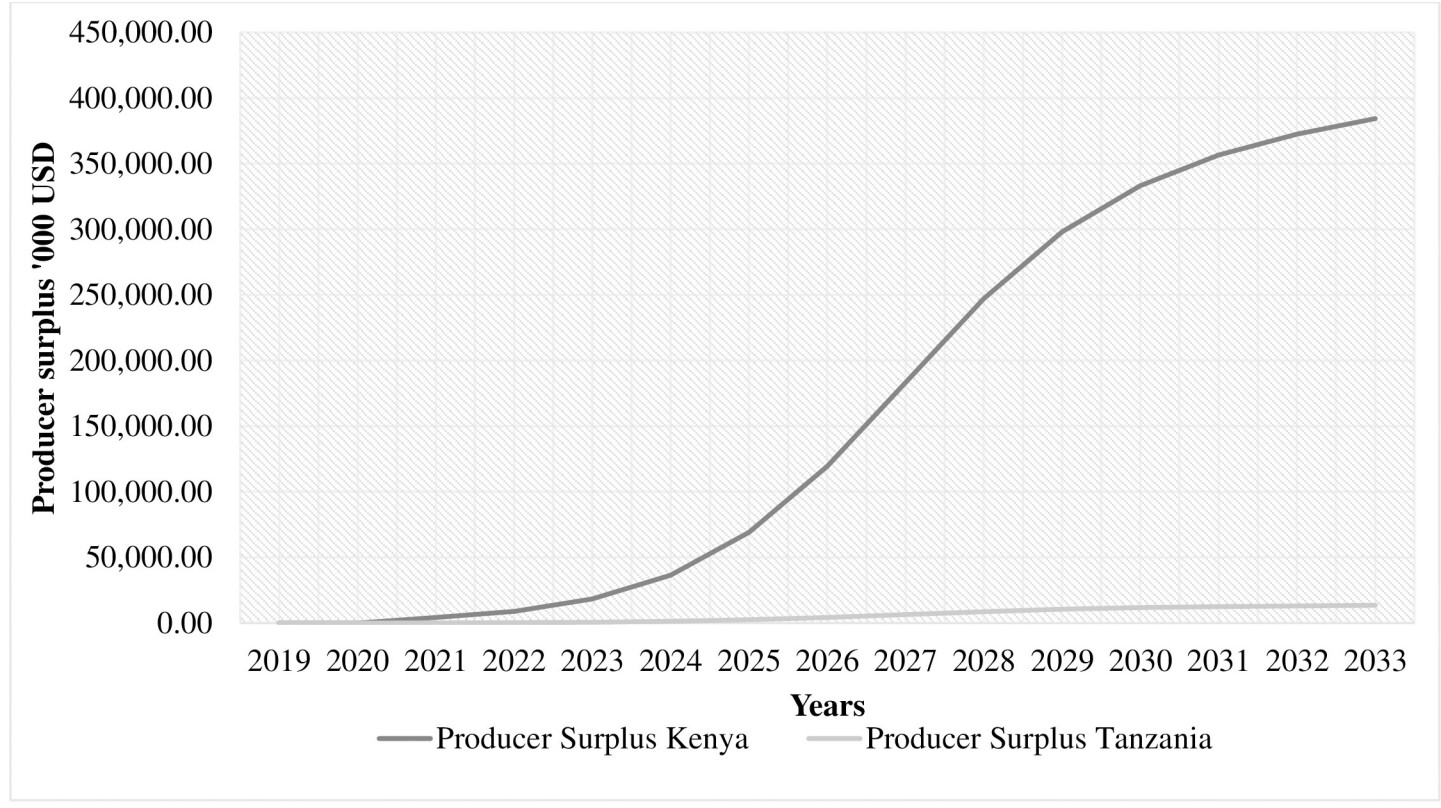

**Fig 2. A comparison of economic surplus over time between Kenya and Tanzania due to Integrated Pest and Pollinator Management intervention in avocado production.**

Though lucrative, the Chinese market requirement to export frozen avocados may lock out many exporters due to the capital-intensive investment in cold chains. Months after Kenya signed the avocado export agreement with China, only one out of over 100 companies has met the requirements for exporting the products to the emerging superpower [48]. The stringent export requirement to China market was facilitated by the need to control quarantine pests such as *Bacrocela Dorsalis*. Similarly, in Tanzania, Africado Ltd. has been pursuing the government to fast-track the export protocol with China to pave the way for local avocado exporters to access this niche market. Sustainable pest management strategies must be put in place for the two nations and the rest of the African countries to penetrate these niche markets. Results show that IPPM has the potential to increase farmers' income due to increased yield, which can be enhanced through compliance with the phytosanitary rules required by the importing countries.

## 3.2 Potential impact of IPPM on poverty reduction

The potential impact of IPPM strategies on poverty reduction is estimated as illustrated by Eq (7). Integrating IPM with pollinator supplementation in Kenya can lift 10,464 people from poverty per year, translating to 156,960 poor people escaping poverty over the simulated period of 15 years. In Tanzania, the number of people estimated to escape poverty due to IPPM intervention is 1,255/year (or 18,825 people over 15 years). In both countries, a majority of poor people live in rural areas where avocadoes are grown, therefore the estimated number of people escaping poverty is a good representation of the expected impact in both economies.

Furthermore, IPPM could have other positive externalities such as effective pollination spill-over effects on the yield of other crops, translating to more gross domestic product for the economy. The spillover effects are however beyond the scope of this study but should be considered for future *ex-post* impact evaluation of IPPM. Besides, other benefits such as beekeeping proceeds from the sale of honey and other products are not quantified, the number of people escaping poverty due to IPPM intervention is therefore conservative. The findings suggest that the adoption of sustainable pest management strategies and enhancing pollinators' population through beekeeping has an instrumental role in economic growth and poverty reduction in Kenya and Tanzania.

### 3.3 Sensitivity analysis

To understand the robustness of model results, we conducted a sensitivity analysis for key parameters to ascertain the effect on results in both extreme pessimistic and optimistic scenarios. We focused our sensitivity analysis on IPPM intervention, the central focus of this study. The *ex-ante* analysis is sensitive to price elasticities of demand and supply, expected yield increase, cost reduction, probability of research success, and maximum adoption rates [24,26]. These parameters were adjusted by -50% and +50%. The results are shown in Table 5. Price elasticity of demand was not expected to influence the economic surplus in a small open economy and our sensitivity analysis conformed to this assumption. The expected yield gain was the most sensitive in both countries, resulting in a significant positive change in IRR, although the figure remained higher than the opportunity cost of capital in both countries. Other variables such as real discount rate and maximum adoption suggest that significant changes in the macro-economic environment (for instance through monetary policies) could enhance farmers' livelihood by acting as an incentive for the adoption of sustainable farming practices.

**Table 5. Sensitivity analysis of key parameters for the economic surplus model.**

| Parameters | Kenya | | | Tanzania | | |
|---|---|---|---|---|---|---|
| | TS ('Million US$) | B/C | IRR (%) | TS ('Million US$) | B/C | IRR (%) |
| *Base values* | *0.996* | *12.9* | *76.3* | *0.026* | *33.9* | *81.3* |
| Yield gain 50% | | | | | | |
| 0.5 | 1.878 | 24.3 | 99.5 | 0.048 | 62.9 | 103.2 |
| -0.5 | 0.171 | 2.2 | 28.8 | 0.005 | 5.9 | 39.2 |
| Cost reduction of 30% | | | | | | |
| 0.5 | 1.229 | 15.9 | 83.5 | 0.032 | 41.7 | 88.1 |
| -0.5 | 0.768 | 10.0 | 68.1 | 0.020 | 26.3 | 73.6 |
| Interest rate (8% Ke; 11% Tz) | | | | | | |
| 0.5 | 0.662 | 10.6 | 76.3 | 0.015 | 20.9 | 81.3 |
| -0.5 | 1.535 | 15.5 | 76.3 | 0.046 | 57.4 | 81.3 |
| Price supply elasticity 0.8 | | | | | | |
| 0.5 | 1.032 | 13.3 | 76.9 | 0.027 | 34.8 | 81.6 |
| -0.5 | 0.959 | 12.4 | 75.7 | 0.025 | 33.1 | 81.0 |
| Probability of success (70% Ke; 60% Tz)) | | | | | | |
| 100% | 1.467 | 19.0 | 89.9 | 0.045 | 58.4 | 100.3 |
| 35% (30%) | 0.480 | 6.2 | 54.4 | 0.013 | 16.6 | 61.1 |
| Max adoption level (85.9% Ke; 68.3% Tz) | | | | | | |
| 100% | 1.173 | 15.2 | 81.9 | 0.039 | 50.8 | 95.0 |
| 35% | 0.388 | 5.1 | 48.7 | 0.012 | 17.0 | 61.7 |

*Note*: TS is Total Surplus; B/C is the benefit over cost; IRR is the internal rate of return; Ke and Tz stand for Kenya and Tanzania respectively.

Furthermore, policies aimed at raising the adoption levels of IPPM such as subsidies for IPM products, and farmer training could benefit the economy and empower the livelihood of many small-scale farmers who dominate the avocado value chain.

## 4. Conclusion and policy recommendations

The adoption of agricultural innovations is important for increasing farm productivity, and income, and sustaining ecosystem services that support livelihoods. Past studies document the impact of adopting sustainable pest management practices such as IPM and pollination services in isolation. In this study, we first assessed the potential (*ex-ante*) economic benefits of integrating IPM and pollination supplementation, an approach referred to as IPPM; next, we estimated the welfare (poverty) effects resulting from these benefits using a case of avocado production in Kenya and Tanzania. Our analysis utilized the economic surplus model and a combination of primary and secondary data.

Our results show that adopting IPPM could generate a significant economic surplus with the potential to reduce poverty. Simulated for 15 years, IPPM would generate an annual net present value (NPV) of US$ 66 million, and US$ 1.7 million in Kenya and Tanzania, respectively. The positive internal rate of return (IRR) further shows the feasibility of IPPM innovation. The generated economic benefits from IPPM innovation are estimated to lift out of poverty 10,464 and 1,255 people annually in Kenya and Tanzania respectively. This suggests the need to encourage farmers to integrate sustainable pests management practices (IPM) and conservation of pollinators particularly those growing pollinator-dependent crops to improve their agricultural productivity and their welfare.

Further, the successful integration of IPPM into the avocado farming system translates to potential positive health and environmental impacts. With reduced use of pesticides, more pollinators included in the ecosystem and higher yields realized, there would be less pressure to transform the land into an agricultural area resulting in conservation and protection of biodiversity and natural habitats building resilience to climate change adversities. These benefits are however beyond the scope of our analysis and are recommended for future studies.

While our study revealed useful insights into the economic and welfare impact of IPPM, we acknowledge further limitations in our analysis. First, in the long run, both the demand and supply of avocado may become elastic in contrast to our assumption. Second, the successful adoption of IPPM and subsequent impact relies on the implementation of supporting policies, whose evaluation was beyond the scope of this study. Furthermore, while our study demonstrates the significant potential economic and welfare impact of integrating IPM and pollinator supplementation, we didn't solely quantify positive externalities of beekeeping such as income generated from the sale of honey and related products. Our estimates are therefore conservative and an *ex-post* evaluation is, therefore, worth future consideration. Assessment of the spillover effects of pollinator supplementation on overall farm crops productivity would also be an interesting knowledge byte.

Given the substantial potential economic and welfare gains demonstrated in this study, policy efforts that encourage the adoption of IPPM should be enhanced. These also include the implementation of feasible export requirements such as the adoption of IPPM and other sustainable and environmentally friendly pest management practices in place of stringent restrictions such as freezing the fruits whose infrastructure is beyond the reach of the smallholder farmers.

## Supporting information

**S1 Data.**
(ZIP)

## Author Contributions

**Conceptualization:** Charity Wangithi, Beatrice W. Muriithi, Gracious Diiro, Thomas Dubois, Samira Mohamed, Michael G. Lattorff, Benignus V. Ngowi, Elfatih M. Abdel-Rahman, Mariam Adan, Menale Kassie.

**Data curation:** Charity Wangithi, Beatrice W. Muriithi, Gracious Diiro, Mariam Adan, Menale Kassie.

**Formal analysis:** Charity Wangithi, Beatrice W. Muriithi, Menale Kassie.

**Funding acquisition:** Beatrice W. Muriithi, Thomas Dubois, Samira Mohamed, Michael G. Lattorff, Benignus V. Ngowi, Elfatih M. Abdel-Rahman, Menale Kassie.

**Investigation:** Charity Wangithi, Beatrice W. Muriithi, Gracious Diiro, Thomas Dubois, Samira Mohamed, Benignus V. Ngowi, Elfatih M. Abdel-Rahman, Menale Kassie.

**Methodology:** Charity Wangithi, Beatrice W. Muriithi, Gracious Diiro, Thomas Dubois, Mariam Adan, Menale Kassie.

**Project administration:** Charity Wangithi, Beatrice W. Muriithi, Thomas Dubois, Samira Mohamed, Michael G. Lattorff, Benignus V. Ngowi, Elfatih M. Abdel-Rahman, Menale Kassie.

**Resources:** Charity Wangithi, Beatrice W. Muriithi, Thomas Dubois, Samira Mohamed, Michael G. Lattorff, Menale Kassie.

**Supervision:** Beatrice W. Muriithi, Gracious Diiro, Menale Kassie.

**Validation:** Charity Wangithi, Beatrice W. Muriithi, Thomas Dubois, Elfatih M. Abdel-Rahman, Mariam Adan, Menale Kassie.

**Visualization:** Beatrice W. Muriithi, Thomas Dubois, Samira Mohamed, Michael G. Lattorff, Menale Kassie.

**Writing – original draft:** Charity Wangithi, Beatrice W. Muriithi, Gracious Diiro, Thomas Dubois, Michael G. Lattorff, Benignus V. Ngowi, Elfatih M. Abdel-Rahman, Mariam Adan, Menale Kassie.

**Writing – review & editing:** Charity Wangithi, Beatrice W. Muriithi, Gracious Diiro, Thomas Dubois, Samira Mohamed, Michael G. Lattorff, Benignus V. Ngowi, Elfatih M. Abdel-Rahman, Mariam Adan, Menale Kassie.

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
