## [Decision Letter · Decision Letter 0]

7 Apr 2022

PONE-D-22-02323The potential economic effect of Integrating Pests and Pollinator Management Strategies in Avocado Farming in East AfricaPLOS ONE

Dear Dr. Muriithi,

Thank you for submitting your manuscript to PLOS ONE. After careful consideration, we feel that it has merit but does not fully meet PLOS ONE’s publication criteria as it currently stands. Therefore, we invite you to submit a revised version of the manuscript that addresses the points raised during the review process.

We look forward to receiving your revised manuscript.

Kind regards,

Javaid Iqbal, PhD

Academic Editor

PLOS ONE

Journal Requirements:

“This work received financial support from the German Federal Ministry for Economic Cooperation and Development (BMZ) commissioned and administered through the Deutsche Gesellschaft für Internationale Zusammenarbeit (GIZ) Fund for International Agricultural Research (FIA), grant number 17.7860.4–001; the Norwegian Agency for Development Cooperation, the Section for Research, Innovation, and Higher Education, grant number RAF-3058 KEN-18/0005; the Swedish International Development Cooperation Agency (Sida); the Swiss Agency for Development and Cooperation (SDC); the Federal Democratic Republic of Ethiopia; and the Government of the Republic of Kenya. The views expressed herein do not necessarily reflect the official opinion of the donors.”

 “D.T., B.W.M, S.M, & M.G.L received the funds. This work received financial support from the German Federal Ministry for Economic Cooperation and Development (BMZ) commissioned and administered through the Deutsche Gesellschaft für Internationale Zusammenarbeit (GIZ) Fund for International Agricultural Research (FIA), grant number 17.7860.4–001; the Norwegian Agency for Development Cooperation, the Section for Research, Innovation, and Higher Education, grant number RAF-3058 KEN-18/0005; the Swedish International Development Cooperation Agency (Sida); the Swiss Agency for Development and Cooperation (SDC); the Federal Democratic Republic of Ethiopia; and the Government of the Republic of Kenya. The views expressed herein do not necessarily reflect the official opinion of the donors. The funders had no role in study design, data collection and analysis, decision to publish, or preparation of the manuscript.”

Reviewers' comments:

Reviewer's Responses to Questions

**Comments to the Author**

1. Is the manuscript technically sound, and do the data support the conclusions?

Reviewer #1: Partly

Reviewer #2: Yes

Reviewer #3: Partly

2. Has the statistical analysis been performed appropriately and rigorously? 

Reviewer #1: I Don't Know

Reviewer #2: Yes

Reviewer #3: Yes

3. Have the authors made all data underlying the findings in their manuscript fully available?

Reviewer #1: Yes

Reviewer #2: Yes

Reviewer #3: Yes

4. Is the manuscript presented in an intelligible fashion and written in standard English?

Reviewer #1: Yes

Reviewer #2: Yes

Reviewer #3: Yes

5. Review Comments to the Author

Reviewer #1: This is an interesting study on an important emerging topic.

However, I think it need some reorganisation and clarification of terms and key issues. This could also lead to reevaluation of the models used and their underlying assumptions.

See attached pdf

Reviewer #2: Comments to authors

The manuscript is valuable and efficient contribution towards the implementation of novel strategies of Integrating Pests and Pollinator Management. However some suggestions are given below for miner revision.

Title

Is good but needs miner rephrasing. Reduce the use of (in)

Abstract

This portion may be good if you add some methodology at line 22 before results. Results may also be elaborated in numerical terms as you have done for cost benefit ratios etc. At the last part of your abstract must indicate future research corridor missing in your research plan.

Keywords

Good

Introduction

At line 73 you must add review about the similar studies already done on any other such crop and their outcome in the form of mini review. Then you must come to your studies.

Line 80-87 seems parts of your results and discussion. These lines may be replaced with the objectives of your studies.

Material and methods

I am more concerned with your methodology as it lacks an integrated methodology you opted. Although you have given concepts, theories before each methodology for subject specialists but for common readers you must elaborate your methodology in a simple form step by step. You may give a schematic diagram of your methodology what were your first step etc. How you started and then ended give step by step in more simpler and understandable way. How you used different formulas and then applied model and how you validated the results of your model. All this can be done with the help of a schematic diagram showing every step and model application procedure.

Results

Before recommendations results must be expressed in a consolidated form.

Discussion must be with citations of similar studies on any other crop if any.

References

References must be according to the format of the journal. This part need the attention of the authors

General Comments

Manuscript is well written. Authors must follow journals format and miner grammatical and language corrections.

Reviewer #3: Article lacks some important details to be fully understand. In addition, it is not well highlighted the importance of this study, and how the finds could be used in future. Please take into account that many sentences need to be rephrased, in my opinion. Moreover, the results should be further developed in the discussion and improve the discussion portion. Some suggested changes as an example are in the comments portion to revise and improve the manuscript. There are many sentences throughout the manuscript which is hard to understand. The present form of draft required a lot of corrections. Please find the comments and suggested corrections. Complete editing corrections, journal-style format, use of abbreviation and missing information should be maintained.

Title: Revise the title of the paper

Abstract: Revision of the abstract is required.

Line 20-22: The authors mentioned “aims to evaluate the potential impact of IPPM elements…? which elements?

Line 22-23: Our results show that the potential economic gain from the adoption of IPPM? What kind of adoption? especially in pest management?

Line 26-29: Restructure the sentence

Rewrite the abstract portion as this is not understandable. The English writing is so confusing to read, it is therefore suggested to rewrite the abstract. English words, vocabulary, journal-style format, word spacing and use correct abbreviations are recommended

Introduction:

Line 44: Restructure the sentence

Line 52-54: Add some relevant references.

Line 69-71: Previous studies have mainly evaluated the economic benefits of IPM technologies and beekeeping in isolation: Add some references to support the statement? which studies?

Line 80-84: The results show significant potential gains……….persons per year out of poverty in Kenya and Tanzania, respectively. This is result portion; add this portion in the results of the study.

Line84-87, 115-116: Restructure the sentence. These sentences are more confusing to understand

Line 131-151: mention the reference of the formulas used in the draft

Line 179-181: These include Murang’a County in Kenya and Kilimanjaro 180 Region in Tanzania…… accounting 181 for 46.9% of the total value of avocado production in the country” Restructure the sentence as this is confusing.

What kind of insects and diseases are present in the avocado crop and what other methods farmer utilized to avoid the chemical control and their economic impact?

Put the survey files in supplementary data as annexures

Line 214-216: Restructure the sentence as this is confusing to understand.

Data sources, parameter estimation, and assumptions” The draft under this sub-headline is too lengthy. Please concise the draft of this subheading

Please add latest reference. Please also see the author’s guidelines of the journal about reference writing. Please mention the research question of why you did this study?

Conceptual model and estimation methods/Materials and Methods:

MM portion needs revision. The sentences are written carelessly; please restructure the sentences and editing used during the study.

Data analysis. Restructure the sentences and add references utilized during the analysis

Results:

The economic analysis of integrating IPM with pollinator supplementation over a 15-year …….which 15 years restructure the sentence as this is confusing (mention the years e.g 2022-2037).

Line 254-260: The authors mentioned the economic benefits from the studied assumptions however they didn’t mention the alternative methods to control insect pests and their economic impact. Please mention this in discussion portion.

The economic comparison of the other alternative crop protection methods to reduce the chemical control in the studied crop area is missing in the discussion portion?

The results portion needs revision. The result portion needs restructuring of the sentences as in the present form the English of the results is not understandable. It is therefore recommended that the authors read to improve the sentences after suggested corrections and before submission.

Discussion:

The findings suggest that the adoption of sustainable pest management strategies and enhancing pollinators' biodiversity through beekeeping has an instrumental role in economic growth and against poverty reduction in Kenya and Tanzania………. What kind of sustainable PMSs?

The discussion portion is very weak and without the discussion of the study conducted. Please add the necessary discussion in the manuscript with the latest reference of the study conducted. Add some latest references in the discussion portion.

Groups of references can be listed either first alphabetically, then chronologically, or vice versa. But you need to be consistent throughout the text. Please correct this in the whole manuscript.

Conclusion:

Conclusion and policy recommendations portion is too lengthy draft and these portions should be written separately in more concise form without the repetition of the information. The conclusion portion is not fine in the present form. Please restructure the conclusion portion by adding the relevant conclusions, and also add the future implication of the present study. Please rewrite in a more concise form.

Figures:

Please also take care of the formatting utilized; there should be uniform formatting throughout the figures. Please also correct the titles by adding the used abbreviations of the figures and formatting according to the author’s instruction of the journal

Tables:

Please correct the titles numbers of the tables in the manuscript.

References: Follow the journal style formatting?

Please add the latest references in the whole manuscript and in literature cited portion

Please double-check for typos and inconsistencies in Journal style/formatting as, among others, missing italics, missing information

---

## [Author Response · Author response to Decision Letter 0]

27 May 2022

Dear reviewers, 

We thank you for your positive feedback, for taking the time, and for providing insightful comments that further improve quality of the paper. We have attempted to address your inputs/comments in the revised version. Below is a table that lists your comments and suggestions and corresponding responses.

Responses to Reviewer #1

1a.This is an interesting study on an important emerging topic. However, I think it need some reorganisation and clarification of terms and key issues. This could also lead to reevaluation of the models used and their underlying assumptions. See attached pdf

Thank you for the extensive review and feedback, We have carefully addressed all your suggestions, and we hope the manuscript now reads better

1b. This is a potential important study on benefits of IPM management and with the inclusion of pollination services the IPPM managements. However, there are some unclarities that makes it hard to evaluate where the assumptions are based on and the definition of some key terms. Especially, the IPPM needs a clear definition of what is added to the IPM. As is now, it is hard to evaluate, as key terms seems to be intermingled or not well defined.

Thank you for the positive feedback on the contribution of our paper. We have edited the definition of IPPM and hope it is now clearer (see lines 47-60, page 3)

Specific comments 

2. Introduction: Line 60-64: Would be good to have a definition of what the authors mean by IPM and IPPM as this can potentially vary between systems and regions. For examples, I would not say adding managed bees to a system makes it a IPPM-system. In comparison, adding bought natural enemies to a crop does not makes the system a IPM-system. Why does the addition of honeybee hives deserves to be called IPPM?

We have edited this section and hope the definition of IPPM is now clear 

3:Conceptual models and estimation methods: 

a.Line 91: Are you sure that this statement is true? For example, farmers can lack information to base the decision to convert to IPPM. Or there might be a investment cost that is to high for some farmers, even if the practice will generate income in long term. 

Thank you for this observation. We have edited the sentence to have more clarity. By economically rational, the farmers are assumed to evaluate the cost and benefits of the technology and only adopt it if they derive any utility from IPPM, i.e. benefits outweigh the costs 

b. Line 111-113: Is this not dependent on whether growers have access to markets both locally and export markets and or distributors? So perhaps growers would sell on local markets if they are more accessible or vice versa. 

Thank you for this positive question. Yes, market access is an important determinant of where farmers would sell their produce. Following a trend of the avocado market in Kenya, Africa, and globally, the export market offers better and more competitive prices. Besides, the local market maybe not be sufficient for the local production; to get foreign earnings, farmers and the government has continuously created a business environment to tap into the increasingly global market niche. As conceptualized in this study, the economic benefits will stream from producer surplus with possible shrink or rise of the domestic market, which we have not quantified due to data limitations. 

c. Line 124-127: Where does this expectation come from? Do you have any references? I would not expect the effect to be direct, especially not for biological control, as it can take some time to increase populations of natural enemies and pollinators. If you support pollination with managed pollinators this will be quicker of course.

The concern is valid. As discussed in the previous paragraph on the impact of IPM, the direct effect of IPPM is attributed to both the use of sustainable pest management practices (IPM) and enhanced yield through pollinator supplementation 

The benefits of the IPM components have been documented in previous studies (see for example Muriithi et al., 2016; Midingoyi et al., 2019). These studies have also been cited in our manuscript 

d. Line 161: Why is the equations jump from 7 to 9? Is there a nr 8 that is missing or is that just a skip of numbers?

This was an oversight, thank you for your for bringing it to our attention

4: Data sources, parameter estimation and assumptions: 

a)Line 183-186: Here you state that there are three options, IPM, pollinator supplementation and

IPPM. However in line 60-62 you state that the IPPM-package integrates IPM with manage pollinators. So if you chose 

IPPM does it include both IPM and manage pollinators? Could you clarify this. Again,

what is the definition of IPPM here?

You are right. During the data collection, we explained the benefits and costs of using IPM alone, supplementation pollination alone, and integrating the two (i.e. IPPM). We then asked the farmers to choose either of the three options that they were willing to adopt. This is the data that we use to derive the adoption rates that we utilize in the analysis of this paper. We then simulated the benefits of each in isolation and integrated them to validate if integrating IPM with pollinators would result in better economic returns. IPPM as described in the introduction, is the Integrated Pest and Pollinator management, i.e. sustainable pest management (i.e. minimal use of chemical pesticides) while enhancing the pollinator population by encouraging the adoption of managed bees (see lines 47-60, page 3)

b) Line 237-239: It seems you take the 20.7% value from the Sagwe et al. 2021 paper or did you

calculate the current pollination deficit? Is taking figures from Sagwe et al., 2021 reliable as you have

farmers in both Kenya and Tanzania? Do you include only smallholder farmers?

This is a valid concern. Yes, we adopted Sagwe et al. 2021. While the author’s results are based in Kenya, the agro-ecological and socio-characteristics of avocado farmers in Kenya and Tanzania are similar. Besides, this is the only most recent study on pollinators in avocado production in East Africa

5: Results and Discussion

a)Line 303?: Line numbering stopped so I had to guess The note March 2022ˆ1 should be used

as a regular reference instead.

We are sorry about this error. The reference is based on a media publication hence its placement as a footnote. However, if the journal’s editor guides otherwise, we are happy to shift it to the regular references 

b)Mid sentences, sec. 4.2: Now you equal IPPM with beekeeping again, but wasn’t managed bees a separate thing from IPPM?

Since the benefits of “pollination” and “IPM” in isolation have been documented before, the gap that this study seeks to address is the integration of the two. In our study area, IPM was introduced as well as managed beekeeping to supplement the existing pollination services. Our focus in Section 4.2, therefore, is the impact of IPPM (IPM + pollination supplementation) on poverty reduction

c) Last sentence, Sec. 4.2: How does beekeeping increase pollinator diversity?

Thank you for your concern. This was a mistake, it’s a pollinator population and not diversity.

b)In general I think you need to discuss the sensitivity analysis a bit more as it was very sensitive to the yield gain. It seems very unsure how big that will be as it is currently based on a secondary report from small scale farmers in one of the countries.

We have added a sentence on the sensitivity of the yield gain variable, hope it's better.

6. Conclusion: I think you would benefit by including your mayor findings and how it impact farmers in the end of the conclusion as a final statement.

Thank you for this suggestion. We have highlighted the major findings (see paragraph 2) and their implications for farmers (see paragraphs 2 & 3)

7: Tables

a)Table 2 In the Note it says PS, CS, PS, I guess the last PS should be TS.

Thank you for the correction. 

b) Table 2 and 3 If CS is always zero, is it not sufficient to say that in the Table text and then remove the CS and TS columns?

This is a valid observation. In an economic surplus model analysis, the three categories of benefits; - PS, CS, and TS must be presented. As such removing may present missing data which is not the case. We suggest keeping the columns. 

References: 

Muriithi, B. W., Affognon, H. D., Diiro, G. M., Kingori, S. W., Tanga, C. M., Nderitu, P. W., ... & Ekesi, S. (2016). Impact assessment of Integrated Pest Management (IPM) strategy for suppression of mango-infesting fruit flies in Kenya. Crop Protection, 81, 20-29.

Midingoyi, S. K. G., Kassie, M., Muriithi, B., Diiro, G., & Ekesi, S. (2019). Do farmers and the environment benefit from adopting integrated pest management practices? Evidence from Kenya. Journal of Agricultural Economics, 70(2), 452-470.

Reviewer #2

1.The manuscript is valuable and efficient contribution towards the implementation of novel strategies of Integrating Pests and Pollinator Management. However some suggestions are given below for miner revision.

Thank you for the positive feedback. 

Specific comments 

2. Title : Is good but needs minor rephrasing. Reduce the use of (in)

Thank you we have rephrased to read; Synergies of Integrated Pests and Pollinator management in avocado farming in East Africa; An ex-ante economic analysis 

3. Abstract: This portion may be good if you add some methodology at line 22 before results. Results may also be elaborated in numerical terms as you have done for cost benefit ratios etc. At the last part of your abstract must indicate future research corridor missing in your research plan.

Thank you; We have added a sentence on the methodology; 

The information on the benefit: cost ratio is also captured. 

For future research, we have followed the journal guidelines. The papers we received captured this information under the conclusion. See the second last paragraph in the conclusion section.

4: Keywords: Good

Thank you 

5. Introduction:

a) At line 73 you must add review about the similar studies already done on any other such crop and their outcome in the form of mini review. Then you must come to your studies.

In the context of Africa, there are no other studies that we are aware of on the potential economic impact of IPPM on any other crops; however, studies on the impact of IPM and pollinators, each studied alone has been cited in the earlier lines of this paragraph.

b)Line 80-87 seems parts of your results and discussion. These lines may be replaced with the objectives of your studies

We have shifted this to results however we borrowed the idea of presenting in brief the results in the introduction section from other related journal papers. 

6.Material and methods: I am more concerned with your methodology as it lacks an integrated methodology you opted. Although you have given concepts, theories before each methodology for subject specialists but for common readers you must elaborate your methodology in a simple form step by step. You may give a schematic diagram of your methodology what were your first step etc. How you started and then ended give step by step in more simpler and understandable way. How you used different formulas and then applied model and how you validated the results of your model. All this can be done with the help of a schematic diagram showing every step and model application procedure. 

Thank you for the suggestion. 

We adopt a common economic model that is widely used and elaborated in the cited references. However, we recognize this may not be common for a non-expert reader. For this, we include “Figure 1: Conceptual framework” which connects the conceptualization and applied method. The framework shows how the economic surplus is derived with equations, and how the variables interact to cause a shift in the economy. Adding an extra diagram (schematic) as you suggested will duplicate Figure 1. 

7: Results: Before recommendations results must be expressed in a consolidated form. Discussion must be with citations of similar studies on any other crop if any; 

Thank you for this observation. Although we recognize recommended journal format of separating the “Results and “Discussion” sections, we request to keep the two together since there are very few studies on IPPM in Africa to correlate to our work. We have however added a few citations on IPM and pollinator’s studies in isolation. Thank you 

8: References: References must be according to the format of the journal. This part need the attention of the authors

We have carefully edited the references following the journal requirements. Thank you

9: General Comments:Manuscript is well written. Authors must follow journals format and miner grammatical and language corrections.

Thank you for the extensive review. We have carefully considered your suggestions and hope the manuscript reads better

Reviewer #3

1: Article lacks some important details to be fully understand. In addition, it is not well highlighted the importance of this study, and how the findings could be used in future. Please take into account that many sentences need to be rephrased, in my opinion. Moreover, the results should be further developed in the discussion and improve the discussion portion. 

Some suggested changes as an example are in the comments portion to revise and improve the manuscript. There are many sentences throughout the manuscript which is hard to understand. The present form of draft required a lot of corrections. Please find the comments and suggested corrections. Complete editing corrections, journal-style format, use of abbreviation and missing information should be maintained.

Thank you for the extensive review of our manuscript. We have tried to improve the overall flow and sentence structure. We do hope with your suggestions the manuscript now reads better.

The importance of the study is highlighted in the fourth paragraph of the introduction 

Detailed specific comments

Detailed comments 

2: Title: Revise the title of the paper

Revised to read as follows

“Synergies of Integrated Pests and Pollinator management in avocado farming in East Africa; An ex-ante economic analysis”

3.Abstract: 

a) Revision of the abstract is required.

Revised

b) Line 20-22: The authors mentioned “aims to evaluate the potential impact of IPPM elements…? which elements?

We have removed “elements”

c) Line 22-23: Our results show that the potential economic gain from the adoption of IPPM? What kind of adoption? especially in pest management?

We are not sure we understand your concern. However, we equate Adoption to “use”; i.e. the literal meaning of “adoption” as used in the use of agricultural innovation/ technology literature. 

d) Line 26-29: Restructure the sentence

Rewrite the abstract portion as this is not understandable. The English writing is so confusing to read, it is therefore suggested to rewrite the abstract. English words, vocabulary, journal-style format, word spacing, and use correct abbreviations are recommended

We have read these lines carefully and edited them accordingly. We hope they are now clearer 

4: Introduction:

a)Line 44: Restructure the sentence

We have edited 

b)Line 52-54: Add some relevant references.

Added (11-13)

c)Line 69-71: Previous studies have mainly evaluated the economic benefits of IPM technologies and beekeeping in isolation: Add some references to support the statement? which studies?

 Cited 19- 23 

d)Line 80-84: The results show significant potential gains……….persons per year out of poverty in Kenya and Tanzania, respectively. This is result portion; add this portion in the results of the study.

Although some journals ask for a brief highlight of results in the introduction section, we have moved these sentences to results.

e)Line84-87, 115-116: Restructure the sentence. These sentences are more confusing to understand

We have edited the sentences. We hope they are clear

f)Line 131-151: mention the reference of the formulas used in the draft

Reference for formulas [25-29]

g) Line 179-181: These include Murang’a County in Kenya and Kilimanjaro Region in Tanzania…… accounting for 46.9% of the total value of avocado production in the country” Restructure the sentence as this is confusing.

We have restructured these sentences, we hope they are now clear

h)What kind of insects and diseases are present in the avocado crop and what other methods farmer utilized to avoid the chemical control and their economic impact?

Due to the space limitation of the paper, we did not go into detail on this although we highlighted the key pests in the second paragraph of the introduction. This information is however provided in a different paper (not published yet) from the baseline survey data that assesses the farmer’s knowledge, attitude, and practices in regard to avocado pests and diseases. 

i)put the survey files in supplementary data as annexures

We are not sure which survey files you are referring to here. if data files, we submitted them as per the journal guidelines 

j) Line 214-216: Restructure the sentence as this is confusing to understand.

Rephrased

5: Data sources, parameter estimation, and assumptions” 

a)The draft under this sub-headline is too lengthy. Please concise the draft of this subheading? Please mention the research question of why you did this study?

Thank you for your concern. We request to keep it as such since reducing it will lose the flow of how parameters are derived. We use both primary and secondary data and this section integrates the two for model estimation, hence the elaborate explanation 

b)Please add latest reference. Please also see the author’s guidelines of the journal about reference writing. 

We have addressed this concern. Thank you

This is mentioned earlier in the introduction, third paragraph

6. Conceptual model and estimation methods/Materials and Methods:MM portion needs revision. The sentences are written carelessly; please restructure the sentences and editing used during the study.

We have carefully read and edited where possible; we hope it has improved. 

7. Data analysis. Restructure the sentences and add references utilized during the analysis

We assume you are referring to the “Results and discussion section”. We have carefully edited this section. For the citation, unfortunately, this being the first study on IPPM, the references that we could cite here are limited. It is also for this reason we have mostly used grey literature as you can see from our list of references. 

8. Results:

a)The economic analysis of integrating IPM with pollinator supplementation over a 15-year …….which 15 years restructure the sentence as this is confusing (mention the years e.g 2022-2037).

We have edited as guided. Thank you 

b)Line 254-260: The authors mentioned the economic benefits from the studied assumptions however they didn’t mention the alternative methods to control insect pests and their economic impact. Please mention this in discussion portion.

Thank you for the suggestion, however, we only focus on this approach for this study. The assessment of alternative methods for control of pests and their economic impacts is however beyond the scope of the current study

c)The economic comparison of the other alternative crop protection methods to reduce the chemical control in the studied crop area is missing in the discussion portion?

While we acknowledge this is an important comparison, we lack studies that use a similar methodology to evaluate alternative methods of crop protection (e.g. use of chemical pesticides). 

d)The results portion needs revision. The result portion needs restructuring of the sentences as in the present form the English of the results is not understandable. It is therefore recommended that the authors read to improve the sentences after suggested corrections and before submission.

We have carefully read through this section and improved the grammar and flow.

9. Discussion

a)The findings suggest that the adoption of sustainable pest management strategies and enhancing pollinators' biodiversity through beekeeping has an instrumental role in economic growth and against poverty reduction in Kenya and Tanzania………. What kind of sustainable PMSs?

In the literature on agricultural innovation/technologies, IPM is considered one of the Sustainable pest management strategies due to its minimal use of pesticides while preserving the econ-system. This definition is briefly provided in the third paragraph of the introduction

b)The discussion portion is very weak and without the discussion of the study conducted. Please add the necessary discussion in the manuscript with the latest reference of the study conducted. Add some latest references in the discussion portion.

We are limited to the few studies on IPPM that have been conducted so far and none from Africa. this is the reason for the light discussion and the reason we merged Results and discussions in the same section.

c)Groups of references can be listed either first alphabetically, then chronologically, or vice versa. But you need to be consistent throughout the text. Please correct this in the whole manuscript.

Thank you for this observation. We have edited the references as guided. 

10. Conclusion: Conclusion and policy recommendations portion is too lengthy draft and these portions should be written separately in more concise form without the repetition of the information. The conclusion portion is not fine in the present form. Please restructure the conclusion portion by adding the relevant conclusions, and also add the future implication of the present study. Please rewrite in a more concise form.

Thank you for the feedback. We followed the journal’s guidelines that do not restrict combining conclusions and policy implications. The journal states the following referring to Results, Discussion, Conclusions; “These sections may all be separate or may be combined to create a mixed Results/Discussion section (commonly labeled “Results and Discussion”) or a mixed Discussion/Conclusions section (commonly labeled “Discussion”). These sections may be further divided into subsections, each with a concise subheading, as appropriate. These sections have no word limit, but the language should be clear and concise”

We have however carefully read through the section to ensure no redundancy of the given information. Reducing the text further might compromise the key messages of the paper. 

11. Figures: Please also take care of the formatting utilized; there should be uniform formatting throughout the figures. Please also correct the titles by adding the used abbreviations of the figures and formatting according to the author’s instruction of the journal

We have addressed this, thank you 

12. Tables:Please correct the titles numbers of the tables in the manuscript.

We have corrected it. Thank you 

13. References: Follow the journal style formatting? Please add the latest references in the whole manuscript and in literature cited portion. Please double-check for typos and inconsistencies in Journal style/formatting as, among others, missing italics, missing information

We have carefully read and addressed these concerns. For the latest references, we are limited to very few that correlate with our study.

---

## [Decision Letter · Decision Letter 1]

27 Jun 2022

Synergies of Integrated Pest and Pollinator Management in Avocado Farming in East Africa: An Ex-Ante Economic Analysis

PONE-D-22-02323R1

Dear Dr. Muriithi,

We’re pleased to inform you that your manuscript has been judged scientifically suitable for publication and will be formally accepted for publication once it meets all outstanding technical requirements.

Kind regards,

Javaid Iqbal, PhD

Academic Editor

PLOS ONE

Additional Editor Comments (optional):

Reviewers' comments:

Reviewer's Responses to Questions

**Comments to the Author**

1. If the authors have adequately addressed your comments raised in a previous round of review and you feel that this manuscript is now acceptable for publication, you may indicate that here to bypass the “Comments to the Author” section, enter your conflict of interest statement in the “Confidential to Editor” section, and submit your "Accept" recommendation.

Reviewer #2: All comments have been addressed

Reviewer #3: All comments have been addressed

2. Is the manuscript technically sound, and do the data support the conclusions?

Reviewer #2: Yes

Reviewer #3: Yes

3. Has the statistical analysis been performed appropriately and rigorously? 

Reviewer #2: Yes

Reviewer #3: Yes

4. Have the authors made all data underlying the findings in their manuscript fully available?

Reviewer #2: Yes

Reviewer #3: Yes

5. Is the manuscript presented in an intelligible fashion and written in standard English?

Reviewer #2: Yes

Reviewer #3: Yes

6. Review Comments to the Author

Reviewer #2: Manuscript now is fine and publishable. All appropriate corrections have been addressed by authors. Materials and methods, Results discussions all are now written in fine form.

Reviewer #3: The article number PONE-D-22-02323R1 submitted to PLOS ONE, entitled “Synergies of Integrated Pests and Pollinator management in avocado farming in East Africa; An ex-ante economic analysis” carried good results. After the complete version of the revision the article can be recommended for the publication of this study. Complete editing corrections, journal style format, use of abbreviation, missing information and double spaces may be improved in proof reading versions.

The article is accepted for publication.

7. PLOS authors have the option to publish the peer review history of their article (what does this mean?). If published, this will include your full peer review and any attached files.

Reviewer #2: No

Reviewer #3: No

---

## [Editor Report · Acceptance letter]

1 Jul 2022

PONE-D-22-02323R1 

Synergies of Integrated Pest and Pollinator Management in Avocado Farming in East Africa: An Ex-Ante Economic Analysis 

Dear Dr. Muriithi:

I'm pleased to inform you that your manuscript has been deemed suitable for publication in PLOS ONE. Congratulations! Your manuscript is now with our production department. 

Kind regards, 

on behalf of

Dr. Javaid Iqbal 

Academic Editor

PLOS ONE